# Tedizolid-Cyclodextrin System as Delayed-Release Drug Delivery with Antibacterial Activity

**DOI:** 10.3390/ijms22010115

**Published:** 2020-12-24

**Authors:** Magdalena Paczkowska-Walendowska, Natalia Rosiak, Ewa Tykarska, Katarzyna Michalska, Anita Płazińska, Wojciech Płaziński, Daria Szymanowska, Judyta Cielecka-Piontek

**Affiliations:** 1Department of Pharmacognosy, Faculty of Pharmacy, Poznan University of Medical Sciences, Swiecickiego 4, 61-781 Poznan, Poland; mpaczkowska@ump.edu.pl (M.P.-W.); nrosiak@ump.edu.pl (N.R.); 2Department of Chemical Technology of Drugs, Poznan University of Medical Sciences, Grunwaldzka 6, 60-780 Poznan, Poland; etykarsk@ump.edu.pl; 3Department of Synthetic Drugs, National Medicines Institute, Chelmska 30/34, 00-725 Warsaw, Poland; k.michalska@nil.gov.pl; 4Department of Biopharmacy, Faculty of Pharmacy, Medical University of Lublin, Chodzki 4a, 20-093 Lublin, Poland; anitaplazinska@umlub.pl; 5Jerzy Haber Institute of Catalysis and Surface Chemistry Polish Academy of Sciences, Niezapominajek 8, 30-239 Krakow, Poland; wojtek_plazinski@o2.pl; 6Department of Biotechnology and Food Microbiology, Poznan University of Life Sciences, Wojska Polskiego 48, 60-627 Poznan, Poland; darszy@up.poznan.pl

**Keywords:** tedizolid, cyclodextrin, dissolution, permeability

## Abstract

Progressive increase in bacterial resistance has caused an urgent need to introduce new antibiotics, one of them being oxazolidinones with their representative tedizolid. Despite the broad spectrum of activity of the parent tedizolid, it is characterized by low water solubility, which limits its use. The combination of the active molecule with a multifunctional excipient, which is cyclodextrins, allows preservation of its pharmacological activity and modification of its physicochemical properties. Therefore, the aim of the study was to change the dissolution rate and permeability through the model membrane of tedizolid by formation of solid dispersions with a cyclodextrin. The research included identification of tedizolid-hydroxypropyl-β-cyclodextrin (tedizolid/HP-β-CD) inclusion complex by thermal method (Differential Scanning Colorimetry), spectroscopic methods (powder X-ray diffraction, Fourier-Transform Infrared spectroscopy), and molecular docking. The second part of the research concerned the physicochemical properties (dissolution and permeability) and the biological properties of the system in terms of its microbiological activity. An increase in the dissolution rate was observed in the presence of cyclodextrin, while maintaining a high permeation coefficient and high microbiological activity. The proposed approach is an opportunity to develop drug delivery systems used in the treatment of resistant bacterial infections, in which, in addition to modifying the physicochemical properties caused by cyclodextrin, we observe a favorable change in the pharmacological potential of the bioactives.

## 1. Introduction

Continuous, progressive increase in bacterial resistance has caused an urgent need to introduce effective and safe new antibiotics or antibacterial agents with a completely new mechanism of action that could effectively inhibit the growth of infections caused by drug-resistant bacteria. Unfortunately, over the past 30 years, despite the urgent need to introduce powerful antibiotics into therapy, the US Food and Drug Administration (FDA) has approved only three new classes of antibacterial agents with a distinct mechanism of action compared to previously approved drugs. One of these classes is oxazolidinones, while diarylquinoline (bedaquiline) and cyclic lipopeptides (daptomycin) are the others [1].

Oxazolidinones are a fully synthetic class of antimicrobials agents. The first registered compound from the oxazolidinone class approved for use in clinical practice was linezolid (2000), followed by tedizolid (2014) [2,3]. Tedizolid at the C5 position of the oxazolidinone ring possess hydroxymethyl sidechain (R-isomer) and contains a biaryl pyridylphenyl ring system [4]. Optimization of tedizolid structure by addition of the pyridine and tetrazol rings allows additional interactions with the upper region of the ribosome 50S peptidyltransferase center site, thereby enhancing the interactions with the binding site [5]. Despite increased efficacy against strains with the *cfr* methyltransferase gene responsible for the resistance of staphylococci to linezolid, tedizolid is low soluble in water with a solubility of 0.136 mg/mL [6]. To solve this problem, the hydroxymethyl substituent at the C5 position of the oxazolidinone ring is first converted into a phosphate ester, and then reacted with sodium metoxide to give the phosphate disodium salt [7]. Pharmacokinetic (PK) and pharmacodynamics (PD) properties of tedizolid phosphate allow the administration of convenient once-daily dosing, a 6-day course of therapy, availability of both oral and intravenous routes of administration. Oxazolidinones, such as beta-lactams, including penicillin and penems, but also glycopeptides, macrolides, azalides, and lincosamides are time-dependent antibiotics, meaning that the time that the concentration of a drug remains above the minimum inhibitory concentration (MIC) (T > MIC) is the PK-PD index correlating with efficacy [8]. The phosphate prodrug moiety provides several fundamental advantages. First, the prodrug approach solves the water solubility problem of the parent tedizolid, allowing parenteral administration of the prodrug with excellent aqueous solubility (>50 mg/mL) [7]. Second the prodrug results in improved bioavailability when compared to parent tedizolid with absolute oral bioavailability of 91% [9]. The prodrug approach is a known molecular modification strategy to improve solubility, pharmacokinetic features, and modulate drug toxicology of the parent drugs [10]. In addition to the benefits of prodrugs, we must also bear in mind the limitations of their use. First, the prodrug is inactive prior to metabolism; it becomes active only after biotransformation, which may differ from person to person and depends on the individual metabolic characteristics. Secondly, increasing the amount of phosphate in the human body, administered in the form of a phosphate prodrug, can cause a decrease in calcium levels, which further results in bone weakness [11], and this will be seriously disadvantageous, especially in the case of tedizolid use in off-label indication for the treatment of bone and joint infections [12].

Like 80% of new drugs, selected oxalidinones are poorly water-soluble. Many efforts have been made to improve the dissolution rate of the drugs, including (1) reducing the drug particle size by mechanical micronization/nanosization techniques, (2) using water-soluble carriers, such as cyclodextrins (CDs), to form inclusion complexes, (3) solubilization in surfactant systems, (4) drug encapsulation or entrapment into liposomes, (5) using prodrugs and drug derivatization, (6) manipulation of the solid state of drug substances improve the drug dissolution i.e., by reducing the crystallinity of drug substances through formation of amorphous solid dispersions (ASDs) and (7) creating multi-component systems such as eutectic drugs, co-amorphous compositions and co-crystals formed by co-crystallization of the active pharmaceutical ingredient (API) with other molecules [13,14]. The choice of the solubilization method should take into account the remaining expectations regarding the modification of pharmacological and physicochemical properties. In the case of antibiotics or antibacterial agents, apart from the required bioavailability, it is important to achieve an effective concentration of the antibacterial agent in accordance with the required dosing algorithm, masking the taste and maintaining bactericidal activity.

The encapsulation of the API into cyclodextrin is a solubilization strategy that allows for potentially additional modification benefits. In the case of antibacterial agents, it is important to obtain appropriately rare dosage algorithms while maintaining bactericidal activity, inhibit the development of bacterial resistance, and finally mask the unpleasant taste. The possibility of cyclodextrin interaction with oxazolidinone analogues was confirmed during the enantioseparation of subsequent derivatives by elecromigration techniques. In this scope, single isomers of β-and γ-cyclodextrin derivatives were tested [15,16,17,18]. However, there are no literature data on the combination of tedizolid with cyclodextrins in the context of the assessment of physicochemical changes in the biomolecule.

The use of hydrophilic polymers such as hydroxypropyl methylcellulose (HPMC) and PEO–PPO–PEO triblock copolymers of poly(ethylene oxide) (PEO) and poly(propylene oxide) (PPO) (trade name Pluronic) significantly influenced the solubility of tedizolid, which resulted in increasing and prolonging its release from received systems. Despite the significant effect on the solubility of tedizolid, a decrease in the permeability of the biomolecule was noted due to the increase in the hydrophilicity of the systems. This resulted in no further development of the proposed tedizolid-based systems [19]. In the case of other classes of antibiotics various benefits have been reported when combining them with cyclodextrins. In case of β-lactam derivatives, an increase in bactericidal activity was found after inclusion into cyclodextrin cavities. For example, meropenem/β-cyclodextrin showed significant changes in MIC values (4.0 mg/mL) against *P. aeruginosa* compared to meropenem (8.0 mg/mL), what can indicate changes in the permeability of the outer membrane [20]. Also, significant improvement in water solubility was observed in case of tebipenem pivoxil, where statistically significant increases in dissolution rate were observed in the case of the tebipenem pivoxil/β-cyclodextrin complex [21], cefuroxime axetil, where after 2 min the cefuroxime axetil/hydroxypropyl-β-cyclodextrin inclusion sample was dissolved in 63%, while pure cefuroxime axetil was only dissolved in 21% [22], or cetirizine/β-CD for taste-masking purpose [23]. The use of cyclodextrin-based systems, in addition to affecting the dissolution rate of biomolecules, may also affect the prolonged and/or delayed release of the active from system [24]. The aim of this study was to develop tedizolid-cyclodextrin systems with functionalities important for the optimization of future chemotherapy using this supramolecular complex such as prolonged release, modification of the penetration through biological membranes or the beneficial change of bactericidal activity. In this respect, hydroxypropyl-β-cyclodextrin (HP-β-CD) was chosen as the optimal solution, due to the European Medicines Agency (EMA) document [25] on cyclodextrins used as excipients in medicinal products for human use. Additionally, HP-β-CD are the most versatile cyclic oligosaccharides that can be used in a variety of medicinal products for oral, rectal, dermal, ophthalmic and parenteral administration, and are considered non-toxic if the daily dose is <16 g (270 mg/kg) [26].

## 2. Results and Discussion

Our research hypotheses assumed that combining the parent tedizolid with cyclodextrin would allow for better solubility and thus modify bioavailability, improve bactericidal activity and mask specific taste.

In the first stage of current research, the composition of the tedizolid/cyclodextrin system was optimized. For studied composition HP-β-CD was chosen, due to EMA document [25]. We obtained the best results for the systems in molar ratio of 1:1 via a kneading method. Stoichiometry of complex formation was chosen based on previous literature data, where molar ratio of 1:1 was predominant [27].

The following XRPD, differential scanning calorimetry (DSC) and FT-IR techniques with theoretical calculations were used as methods to confirm the identity of the obtained system. Pure tedizolid, HP-β-CD and their solid system obtained by kneading were characterized using powder X-ray diffraction (PXRD). The PXRD patterns, shown in Figure 1, revealed the crystalline nature of the parent tedizolid, as demonstrated by sharp Bragg peaks at 10.0°, 12.4°, 14.4°, 16.9°, 19.6°, 22.7°, 23.8°, 26.4°, 28.6°, and 29.2° 2θ, while the HP-β-CD starting material has an amorphous nature. The binary system diffractogram is a sum of patterns of each individual phase (pure tedizolid and HP-β-CD).

The DSC thermograms collected for pure components, and of the tedizolid/ HP-β-CD system are presented in Figure 2. The DSC thermograms obtained for the parent tedizolid demonstrated a sharp endothermic peak with a minimum temperature at 205.6 °C, which may be related to the melting point of tedizolid, which is typical for an anhydrous crystalline drug. The DSC traces of HP-β-CD reveal a broad endothermal peak located around 125.6 °C, associated with loss of water. HP-β-CD as an amorphous compound does not exhibit a melting point, as would be observed for crystalline compound. In the case of the tested formulation, two endothermic processes were recorded but reduced in intensity during DSC measurements. The first, with the maximum located at 118.8 °C reflects the water evaporation from the samples, and the second, more significant endothermic peak with a maximum at 200.1 °C corresponds to the melting of tedizolid. The melting point of tedizolid in the HP-β-CD system is lower temperatures due to molecular encapsulation of tedizolid into the HP-β-CD nanocavity.

Based on a comparison of the enthalpy of the parent API fusion and the system with HP-β-CD, the extent of CD inclusion could be estimated. Knowing the molecular weight of all components as well as the weight percent of tedizolid in the formulations, the enthalpy of fusion of ideal systems can be calculated in which there is no inclusion. Comparing these, calculated values, to the experimental data, it was possible to estimate that in the tedizolid/HP-β-CD formulation, 17.52% of tedizolid is trapped inside HP-β-CD.

The Figure 3 shows the FT-IR spectrum for the tedizolid, HP-β-CD and inclusion complex of tedizolid/HP-β-CD. The FT-IR spectra of tedizolid include significant vibrations at 879 cm^−1^ (C-H wagging), 1619 cm^−1^ (C=C stretching + C-C stretching in phenyl ring), 3259 cm^−1^ (symmetric C-H stretching in phenyl ring) in the (Pyridine-3-yl)phenyl-3-fluoro structure, 1223 cm^−1^ (C-O stretching vibrations, 1328 cm^−1^ (C-H wagging), 1410 cm^−1^ (C-H wagging and C-N-C bending vibrations), 1747 cm^−1^ (C=O stretching vibrations) in the 1,3-oxazolidin-2-one structure and 1196 cm^−1^ (C-H wagging in pyridine ring and C-C-N bending vibrations in the (Pyridine-3-yl)phenyl-3-fluoro structure [28]. The FT-IR spectra of HP-β-CD consisted of the prominent absorption band of the 1034 cm^−1^ (C-C stretching vibrations), 1085 cm^−1^ (C-O stretching vibrations), 1158 cm^−1^ (symmetric C-O-C stretching vibrations), 1654 cm^−1^ (H-OH bending vibrations), 2973 cm^−1^ (symmetric and a symmetric C-H stretching vibrations in CH_2_ and CH_3_ groups in in hydroxypropyl parts) and 3411 cm^−1^ (OH stretching vibrations). Furthermore, in the range of 1200–1500 cm^−1^ there are bands associated with bending vibrations of C-H and O-H bonds [29,30,31,32,33].

The changes observed as the absorption spectrum of tedizolid/HP-β-CD indicate the formation of an inclusion complex. The formation of the complex carries the limitations of bending and stretching vibrations of the guest molecules. The literature often indicates changes in the intensity of the mentioned bands and/or their shifts in the recorded absorption spectra [31,34,35]. The broad band corresponding from OH bonds (3411 cm^−1^) in HP-β-CD consistently appears in the tedizolid/HP-β-CD complex. However, it should be noted that it is broaden and shifted towards shorter wavelengths. According to literature, it may be related to the formation of hydrogen bonds between tedizolid and cyclodextrin [29,30,31]. The numerous bands present in the spectrum of pure tedizolid in the range of 950–1200 cm^−1^ in the complex have largely been obscured by the characteristic absorption peaks of HP-β-CD (C-C (1034 cm^−1^), C-O (1084 cm^−1^) and C-O-C (1158 cm^−1^) stretching vibrations). Among them, it could be distinguish bands derived from pure tedizolid: 1017 cm^−1^ (C-O stretching vibration, C-C-N, C-C-C and C-N-C bending vibrations in the 1,3-oxazolidin-2-one structure), 1045 cm^−1^ (C-C stretching vibrations, deformation rings in (pyridine-3-yl)phenyl-3-fluoro structure and O-H wagging in hydroxymethyl structure), 1104 cm^−1^ (C-O stretching vibrations in the 1,3-oxazolidin-2-one structure), 1196 cm^−1^ (C-H wagging in pyridine ring and C-C-N bending vibrations in the (pyridine-3-yl)phenyl-3-fluoro structure). Moreover, the spectrum of the complex shows a decrease in the intensity of the bands 879, 1196, 1223, 1282, 1328, 1380, 1410, 1619, 1747 cm^−1^ characteristic of the structure of 1,3-oxazolidin-2-one and (pyridine-3-yl) phenyl-3-fluoro of tedizolid. These bands are mainly related to the stretching vibrations of the C-O, C-C, C--F bonds and the C-H wagging [28]. These results confirm the obtained docking simulations, which indicate that the 1,3-oxazolidin-2-one structure with the hydroxymethyl group attached to it interacts with hydroxypropyl groups of HP-β-CD.

The binding energies found during docking simulations are equal to 24.7 and 28.2 kJ/mol, for universal force field (UFF) and density functional theory (DFT) structures, respectively. Both the sign of the determined values and their magnitudes suggest a strongly favorable binding mode. Additionally, the same quantity was determined at the DFT level, at the stage of final geometry optimizations, followed by analogous calculation performed for unbound HP-β-CD and tedizolid molecule. By subtracting the corresponding energy terms, the binding energies of order of 32.7 and 37.2 kJ/mol were obtained for UFF and DFT structures, respectively. The favorability of binding was again confirmed.

The analysis of the DFT-optimized tedizolid/HP-β-CD complexes provided an insight into the qualitative pattern of interactions that may be the binding driving force as well as into the preferred spatial arrangement of the bound ligand molecule into the cavity of HP-β-CD. The graphical illustration of the considered structures is given in Figure 4. The two alternative structures of HP-β-CD differ mainly by the orientation of their most flexible parts, i.e., the hydroxypropyl moieties. We decided to consider both DFT and UFF structures due to the fact that in the case of physical systems, the orientation of the substituents is expected not to be conformationally locked but, conversely, they can adopt various geometries with the associated dynamic equilibrium established through conformational rearrangements occurring within several picoseconds. This is likely to be correlated with the coexistence of multiple binding modes, differing mainly by the pattern of interactions of the tedizolid molecule with HP moieties.

Independently of the considered structure, the orientation of the ligand molecule in the binding cavity is roughly the same. The two central aromatic moieties of the ligand are in close contact with the inner cavity of HP-β-CD, created by the glucose moieties. The close proximity of those rings with the aliphatic hydrogen atoms suggest the contribution of the CH-π interactions, characteristic of the carbohydrate binding other biomolecules [36]. At the same time, in the case of UFF structure, the 1,3-oxazolidin-2-one structure and the hydroxymethyl moiety attached to it maintain the contact with the hydroxypropyl groups of HP-β-CD. The contact involves both hydrogen bonding with hydroxyl groups and the contact between nonpolar parts of those groups. The opposite end of the ligand molecule, i.e., tetrazole moiety, exhibit only a looser contact with hydroxymethyl moieties of glucose residues. Such proximity suggests the possibility of the corresponding intermolecular hydrogen bonding, although no such contact type was observed in any of the two considered structures. The slightly different pattern of interactions is exhibited in the case of DFT structure, where the 1,3-oxazolidin-2-one structure does not maintain any close contact with the hydroxypropyl groups, and the binding is limited to the remaining contacts.

Judging from the other studies in which cyclodextrins exhibit a significant molecular flexibility as well as from the current results according to which at least two energetically favorable structures were found, we speculate that the actual binding mode may be a superposition of numerous configurations of tedizolid molecule in the binding cavity, differing mainly in the interaction patterns involving the most conformationally flexible parts of both molecules, i.e., the 1,3-oxazolidin-2-one structure and the hydroxypropyl moieties attached to cyclodextrins. Nevertheless, the main driving force for binding is identified unequivocally and include the contacts between hydrophobic patches of glucose residues located in inner cyclodextrin cavity and the two central aromatic residues of tedizolid moiety. The responsible type of intermolecular forces most likely originates from the CH-π interactions.

Comparisons on differences on XRPD, DSC and FI-IR spectra confirm that for this system in molar ratio of 1:1, it is possible to find intermolecular interactions. Theoretical optimization and analysis of FT-IR spectra detail the molecular domains involved in intermolecular interactions.

The second part of the study focused on the studies of physicochemical (dissolution, permeability through artificial biological membranes) and microbiological activity properties after the interaction between tedizolid and CD. The physicochemical properties of tedizolid systems were evaluated with the use of the high-performance liquid chromatography with a diode-array detector (HPLC-DAD) method developed for the determination of tedizolid. For this purpose, the HPLC-DAD method in gastric juice (pH 1.2) and in phosphate buffer (pH 4.5 and pH 6.8) was validated according to The International Council for Harmonisation of Technical Requirements for Pharmaceuticals for Human Use (ICH) guidelines (Appendix A).

The solubility of tedizolid phosphate is the highest when the drug is fully deprotonated (high pH) and lowest when fully protonated (low pH), with a steep increase in solubility between pH 4.0 and pH 6.0 [37]. The active substance, tedizolid phosphate-free base, has limited aqueous solubility and is stable across a pH range of 3 to 8 [38]. The changes in the dissolution rate of parent tedizolid, as well as tedizolid/HP-β-CD complex, confirm a positive solubility effect of cyclodextrin; however, the presence of HP-β-CD did not change the dissolution profile shape (Figure 5). Calculated *f*_1_ and *f*_2_ values confirmed that the dissolution profile of the tedizolid/HP-β-CD complex is very different from parent tedizolid in three dissolution media (*f*_1_/*f*_2_ values: 27.99/3.17, 98.19/3.57, and 68.89/10.03 at pH 1.2, 4.5 and 6.8, respectively). Interestingly, the speed of the dissolution of the tedizolid/HP-β-CD complex was from twice (at pH 4.5 and 6.8) up to three times higher (pH 1.2) compared with the parent tedizolid. Such different dissolution profiles show the potential of cyclodextrins as substances that significantly modify tedizolid release, especially in the gastric environment. Based on these findings, despite the absence of other oxazolidinones studies, confirm the usefulness of using cyclodextrin to increase the solubility of other sparingly water-soluble antibiotics, e.g., tebipenem pivoxil from the carbapenem group [21].

The ability of tedizolid to penetrate biological membranes under the digestive tract condition was tested using the parallel artificial membrane permeability assay (PAMPA system). Tedizolid, classified to II BCS class, as a poorly water-soluble drug, shows high permeability. As expected, permeability studies using the PAMPA system, through the gastrointestinal barrier, confirmed that the apparent permeability coefficients for the parent tedizolid and its inclusion complex with HP-β-CD were higher than 1 × 10^−6^ cm s^−1^, classifying all of them as high-permeability systems (Figure 6) [39]. It confirmed the previous in vitro studies, where tedizolid has high permeability across Caco-2 monolayers, which suggests that tedizolid may enter into hepatocytes primarily via passive diffusion [40]. Tedizolid phosphate has low membrane permeability and the stomach does not contain phosphatase enzymes, therefore in vitro studies suggest that tedizolid phosphate is dephosphorylated in the intestinal brush border membrane, after which tedizolid enters the enterocytes without accumulation in the intestinal lumen [41]. What is interesting, parent tedizolid permeates faster in the first 120 min. After that time, there was no significant difference in the apparent permeability coefficient for tedizolid and tedizolid/HP-β-CD. This confirms the hydrophilic nature of the complex is not favored by passive penetration. High penetration of the complex at 240 min may be due to the high degree of dissolution of tedizolid and confirms the sense of prolonged use of the drug.

A microbiological analysis to determine the MIC of tedizolid (control) and the active substance enriched with an additive such as HP-β-CD was determined for 4 bacterial species, including reference strains and clinical isolates (Table 1). Bacterial strains were selected for both general infections and acute bacterial skin and skin structure infections (ABSSSI). Tedizolid/HP-β-CD system caused an increase of activity expressed by the decrease of MIC value in the case of *E. faecalis* and *E. faecium*, for both reference strain and clinical isolates. ABSSSI is mainly caused by *Staphylococcus aureus* [42], for which the MIC value was 2 mg L^−1^ for clinical isolates. What is essential, there was no decreased activity after complexation with HP-β-CD. In case of high action against *E. faecalis*, it is possible to conclude that cyclodextrins by blocking porin channels contribute to the efflux effect in bacteria [21]. Moreover, cyclodextrin molecules may have potentially destabilized the outer membrane of the bacteria, which eventually lead to an increase in the diffusion rate of the antibiotics [43].

## 3. Materials and Methods

Tedizolid (purity >98%) were supplied by T&W GROUP (Shanghai, China). Hydroxypropyl-β-cyclodextrin (HP-β-CD) (purity >98%) was obtained from Sigma-Aldrich (Poznan, Poland). UHPLC grade Acetonitrile UHPLC grade was supplied by Merck KGaA (Darmstadt, Germany) and formic acid (100%) by Avantor Performance Materials (Gliwice, Poland). Hydrochloric acid, potassium bromide, and potassium dihydrogen phosphate were obtained from Avantor Performance Materials (Gliwice, Poland). High-quality parent water was prepared by using a Direct-Q 3 UV Merck Millipore purification system (Molsheim, France).

### 3.1. System Preparation

The solid inclusion complex of tedizolid with hydroxypropyl-β-cyclodextrin (HP-β-CD) was obtained by the kneading method/mechanochemical activation method [44]. Starting material as tedizolid and HP-βCD, without any solvent, was kneaded in the same molar ratio 1:1 with continuous stirring for 30 min. The process is known as dry mechanochemical activation. Then, the system was stored at constant ambient humidity in the evacuated chambers at 20 °C.

### 3.2. The Identification of the Systems

Identification of the tedizolid systems was confirmed with regards to results of thermal (changes in DSC thermograms) and spectroscopic (changes in FT-IR spectra, changes in powder X-ray diffraction (PXRD) patterns) studies.

#### 3.2.1. Differential Scanning Colorimetry (DSC)

DSC analysis of all compounds was performed using a DSC 8500 Perkin Elmer equipped with an intercooler system. Indium was used for calibration. Accurately weighed samples were placed in 60 μL sealed cells, and heated at a scanning rate of 10 °C min^−1^ from 25 °C to 250 °C under a nitrogen purge gas with a flow rate of 20 mL min^−1^. Each run was repeated at least twice.

#### 3.2.2. Powder X-ray Diffraction (PXRD)

PXRD analysis was performed at ambient temperature using the Bruker D2 Phaser diffractometer with LynxEye XE-T 1-dim detector and Cu Kα radiation (λ = 1.54056 Å, generator setting: 40 kV and 40 mA). Diffraction data were collected at the 2*θ* scanning range between 5° to 40° with a step size of 0.02° and a counting time of 2 s/step.

#### 3.2.3. FT-IR Spectroscopy

All compounds were obtained separately with IR grade potassium bromide at a ratio of 1:100, and IR pellets were prepared by applying 8 metric tons of pressure in a hydraulic press. The vibrational infrared spectra were measured between 4000 cm^−1^ and 4000 cm^−1^, with an FT-IR Bruker Equinox 55 spectrometer equipped with a Bruker Hyperion 1000 microscope. To analyze changes in the positions and intensity of bands in the experimental spectra of the systems, quantum-chemical calculations were performed based on B3LYP functional and 6-31G(d,p) as a basis set. All calculations were performed using the Gaussian 09 package and the GaussView application.

#### 3.2.4. Molecular Docking

The ligand molecule was prepared by using the Avogadro 1.1.1 software [45] and optimized within the UFF force field [46] (5000 steps, steepest descent algorithm). The structure of HP-β-CD was prepared on the basis of available crystal structure of unfunctionalized β-cyclodextrins, the results of our earlier (unpublished) studies as well as the structural parameters available in the literature [47]. According to the experimental data, the molar weight of HP-β-CD corresponds to a unique substitution pattern, according to which only one of glucose residue remains unfunctionalized. The hydroxymethyl groups were initially oriented in the trans conformation, in agreement with the dominated conformer found in glucose monosaccharide. The 2-HP moieties were oriented in order to facilitate the hydrogen bonding-mediated contact between neighboring units. The resulting structure was optimized within the UFF force field (10,000 steps) and either directly subjected to docking or further optimized at the high-level theory (DFT/B3LYP/6-311G(d,p) [48,49,50]). The flexible, optimized ligand molecule was docked into the two structures of HP-β-CD (denoted, in accordance with the optimization level, as UFF and DFT). The AutoDock Vina software [51] was applied for docking simulations. The procedure of docking was carried out within the cuboid region large enough to accommodate the whole HP-β-CD molecule. All the defaults procedures and algorithms implemented in AutoDock Vina were applied during docking procedure. The complexes exhibiting the most favorable binding energy and were extracted and subjected to geometry optimization at the DFT/B3LYP/6-311G(d,p) level of theory. Both stages of the DFT calculations were carried out within Gaussian09 software (Wallingford, CT, USA) [52].

### 3.3. Studies of Physicochemical Properties of Systems

The changes in systems (parent tedizolid vs. tedizolid/HP-β-CD) concentrations during dissolution and permeability studies by using the HPLC-DAD method were determined. In this order, the HPLC-DAD method was developed and validated. The separation of tedizolid was possible using the liquid chromatography system (Dionex Thermoline Fisher Scientific, Dreieich, Germany) equipped with a high-pressure pump (UltiMate 3000), an autosampler (UltiMate 3000) and a DAD detector (UltiMate 3000) with Chromeleon software version 7.0 from Dionex Thermoline Fisher Scientific (Dreieich, Germany). Separations were performed on a Kinetex-C18 column (100 mm × 2.1 mm, 5.0 μm) (Torrance, CA, USA). The detection of tedizolid was performed using a diode array detector at a wavelength maxima (𝜆_max_) of 300 nm. The mobile phase consisted of a mixture of 0.1% formic acid and acetonitrile (80:20 *v*/*v*) with a mobile phase flow rate of 1.0 mL min^−1^. The column was set at 30 °C.

The changes of tedizolid concentrations were measured in order to determine the differences between pure tedizolid and in systems with cyclodextrin during dissolution and permeability studies.

The HPLC-DAD method was validated according to the International Conference on Harmonization Guidelines. It comprised selectivity, linearity, accuracy, precision, limits of detection (LOD) and quantitation (LOQ) [53].

#### 3.3.1. Dissolution of Systems

Dissolution studies of tedizolid from cyclodextrin systems by using a standard paddle Agilent 708-DS Dissolution Apparatus (Santa Clara, CA, USA) with a 500-mL dissolution medium at 37 ± 0.5 °C and 50 rpm for 240 min were performed. Samples were weighed into gelatine capsules and then placed in the spring in order to prevent flotation of the capsule on the surface of the liquid. As dissolution media, artificial gastric juice at pH 1.2 and phosphate buffer at pH 4.5 and pH 6.8 were used. At appropriate time intervals, dissolution samples (5.0 mL) were collected with the replacement of equal volumes of temperature-equilibrated media and filtered through a 0.45 μm membrane filter. Preparation of all solvents and procedures of determinations were conducted according to the requirements of Pharmacopeia guidelines [54]. The dissolved drug concentration was measured by using the HPLC-DAD method. The similarity of dissolution percentage of tedizolid in different forms (until it achieves plateau) was established based on f_1_ and f_2_ parameters and was defined by the following equations:$$f_{1}=\frac{\sum_{j=1}^{n}\left|R_{j}-T_{j}\right|}{\sum_{j=1}^{n}R_{j}}\times 100\;f_{2}=50\times \log\left({\left(1+\left(\frac{1}{n}\right)\sum_{j=1}^{n}{\left|R_{j}-T_{j}\right|}^{2}\right)}^{-\frac{1}{2}}\times 100\right)$$
in which *n* is the number of withdrawal points, *R*_j_ is the percentage dissolved of reference at time point *t*, and *T*_j_ is the percentage dissolved of test at time point *t*. According to the Moore and Flanner model, the value of 0 for *f*_1_ and the value of 100 for *f*_2_ suggests that the test and reference profiles are identical [55]. Values between 50 and 100 indicate that the dissolution profiles are similar, whereas smaller values imply an increase in dissimilarity between release profiles.

#### 3.3.2. Permeability Studies of Systems

Differences in the gastrointestinal permeability of tedizolid and parent tedizolid with HP-β-CD were investigated by using a PAMPA method (parallel artificial membrane permeability assay). The system consisted of a 96-well microfilter plate and a 96-well filter plate and was divided into two chambers: a donor at the bottom and an acceptor at the top, separated by a 120-μm-thick microfilter disc coated with a 20% (*w*/*v*) dodecane solution of a lecithin mixture (Pion, Inc., Billerica, MA, USA). The donor solution was adjusted to pH 1.2, 4.5, and 6.8. The substances were dissolved in the donor solution. The plates were put together and incubated at 37 °C for 240 min in a humidity-saturated atmosphere. The concentrations of substances in the donor and acceptor compartments were determined by using HPLC-DAD method.

The apparent permeability coefficient (*P*_app_) was calculated from the following equation:$$P_{app}=\frac{-ln\left(1-\frac{C_{A}}{C_{equilibrium}}\right)}{S\times \left(\frac{1}{V_{D}}+\frac{1}{V_{A}}\right)\times t}$$
where *V*_D_—donor volume, *V*_A_—acceptor volume, *C_equilibrium_*—equilibrium concentration $C_{equilibrium}=\frac{C_{D}\times V_{D}+C_{A}\times V_{A}}{V_{D}+V_{A}}$, *C*_D_—donor concentration, *C*_A_—acceptor concentration, *S*—membrane area, *t*—incubation time (in seconds).

To verify that *P*_app_ determined for permeability was statistically different, an analysis of variance (ANOVA) test was used. Compounds with *P*_app_ < 1 × 10^−6^ cm s^−1^ are classified as low-permeable and those with *P*_app_ > 1 × 10^−6^ cm s^−1^ as high-permeable compounds [39].

#### 3.3.3. Studies of Bactericidal Activity of Tedizolid in Tedizolid/HP-β-CD Inclusion Complex

All test samples (tedizolid, HP-β-CD and tedizolid/HP-β-CD) were dissolved in 1.0 mL of dimethyl sulfoxide (DMSO) and mixed until fully dissolved. From so-obtained stock solutions at a concentration of 100 mg mL^−1^, a series of dilutions in the concentration range 0.5–15 mg mL^−1^ in Antibiotic Broth medium (Merck) were prepared. For each l.0 mL of dilution, 0.1 mL of 18-h-old liquid culture of standard strains, diluted 1:10,000 in the same Antibiotic Broth medium, was added (the number of added cells was approximately 103 in 0.1 mL). The samples were incubated at 37 °C for 24 h. After this time, all dilutions were inoculated on solid Antibiotic Agar. After a further 24 h, the lowest concentration of sample dilutions, which completely inhibited the reference strain growth, was marked as Minimal Inhibitory Concentration.

## 4. Conclusions

Tedizolid with the HP-β-CD system has been proven to be an effective delivery system of antibacterial agent. Importantly, the tedizolid exhibited a delayed-release profile from the prepared delivery system. Additional benefits of the prepared system include the increase in tedizolid permeability as a consequence of its better dissolution over a longer period of time and an increase in the bactericidal effect. The prospect of using tedizolid/HP-β-CD system as a delayed-release drug delivery is important due to the possibility for optimizing chemotherapy in clinically serious infections.

## Figures and Tables

**Figure 1 ijms-22-00115-f001:**
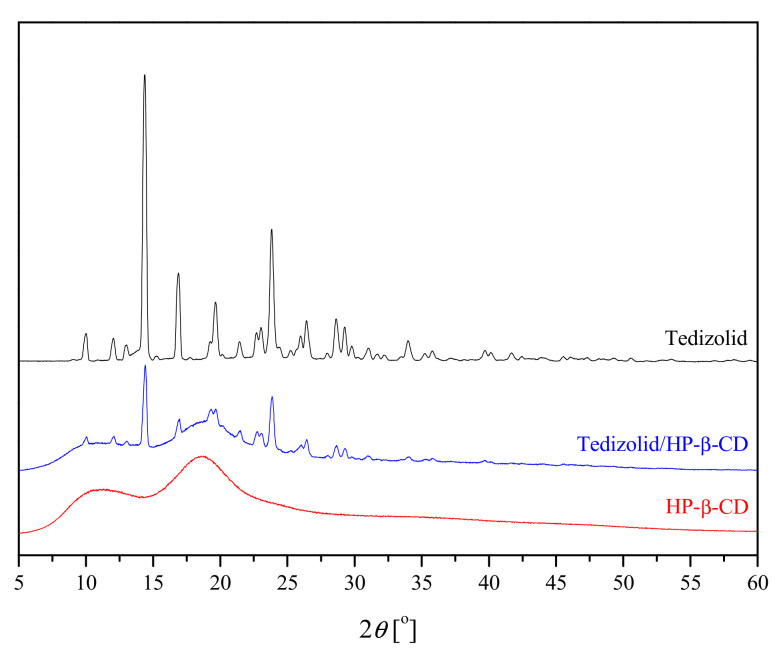
X-ray diffractograms of tedizolid, HP-β-CD and tedizolid/HP-β-CD complex.

**Figure 2 ijms-22-00115-f002:**
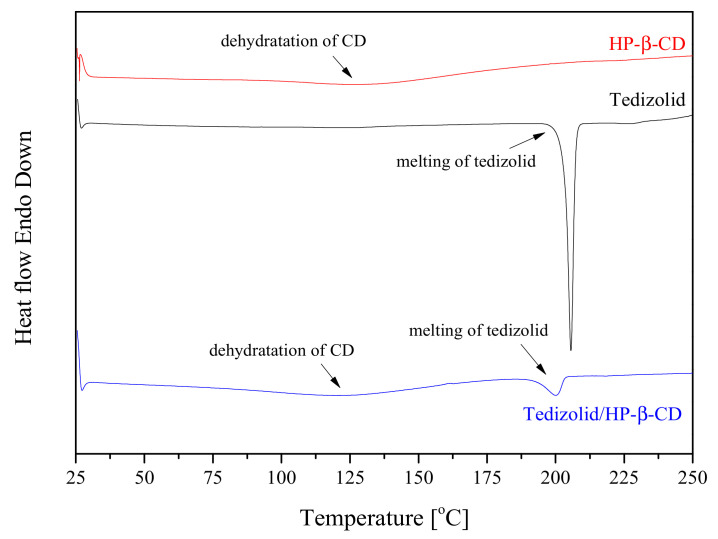
Thermograms of tedizolid, HP-β-CD and tedizolid/HP-β-CD complex.

**Figure 3 ijms-22-00115-f003:**
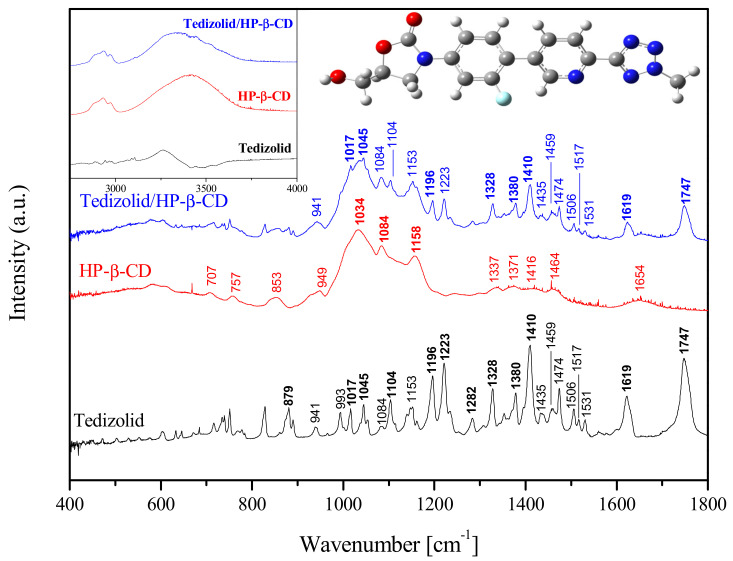
FT-IR spectra of tedizolid, HP-β-CD and tedizolid/HP-β-CD complex.

**Figure 4 ijms-22-00115-f004:**
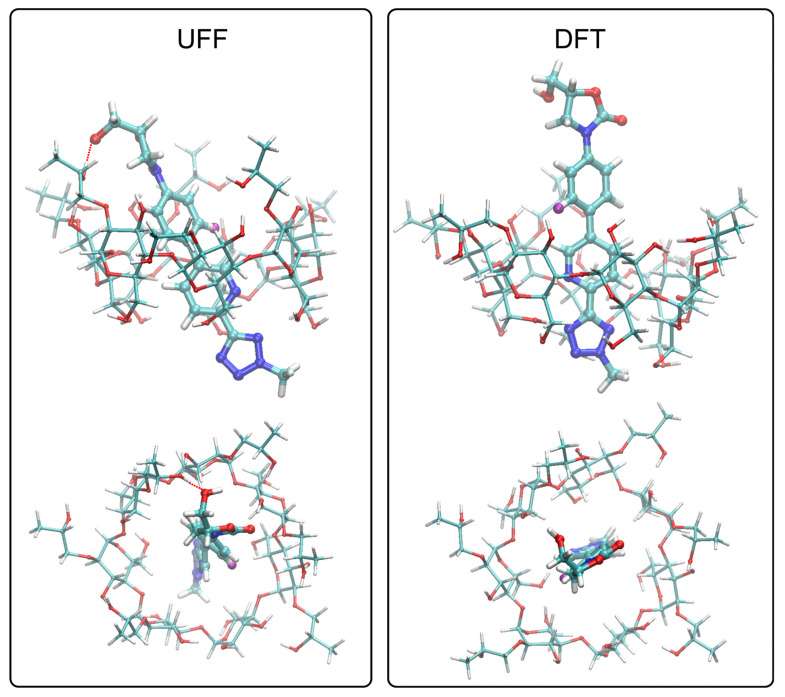
The conformation of the tedizolid/HP-β-CD.

**Figure 5 ijms-22-00115-f005:**
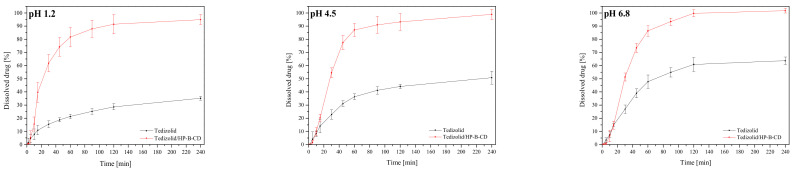
Dissolution profiles of tedizolid from the powder CD complex at pH 1.2/4.5/6.8.

**Figure 6 ijms-22-00115-f006:**
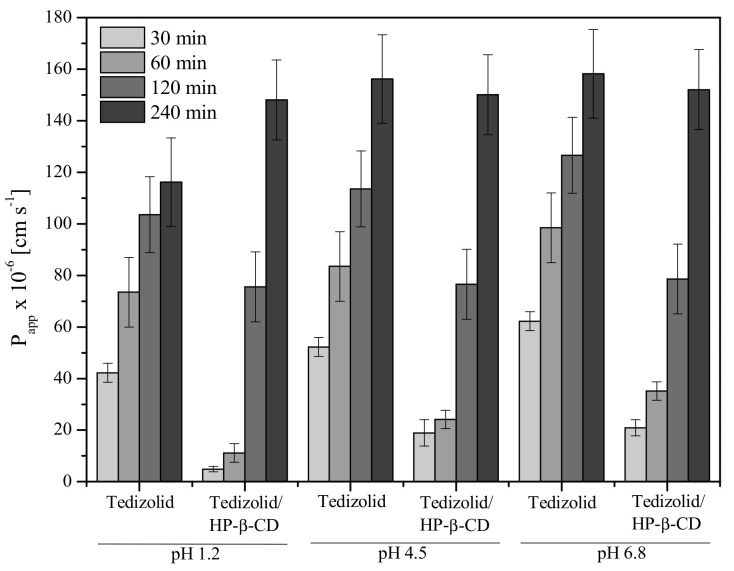
The apparent permeability coefficient values of the tedizolid and tedizolid/HP-β-CD system at pH 1.2/4.5/6.8.

**Table 1 ijms-22-00115-t001:** MIC value for tedizolid and tedizolid/HP-β-CD.

	Tedizolid	Tedizolid/HP-β-CD
	MIC [mg L^−1^]	
*Enterococcus faecalis* ATTC 29212	0.5 ± 0.1	0.25 ± 0.1 ↓
*E. faecalis* clinical isolates	1 ± 0.2	0.25 ± 0.1 ↓
*Enterococcus faecium* ATCC 27270	0.5 ± 0.1	0.25 ± 0.1 ↓
*E. faecium* clinical isolates	1 ± 0.2	0.5 ± 0.1 ↓
*Staphylococcus aureus* ATCC 25923	0.5 ± 0.1	0.5 ± 0.1
*S. aureus* clinical isolates	2 ± 0.2	2 ± 0.2
*Staphylococcus epidermidis* ATCC 12228	1 ± 0.2	1 ± 0.2
*S. epidermidis* clinical isolates	2 ± 0.2	2 ± 0.2

↓—decrease in MIC value.

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
