# Peer review of "Tedizolid-Cyclodextrin System as Delayed-Release Drug Delivery with Antibacterial Activity"

_ijms, 2020, doi:10.3390/ijms22010115_

Round 1

Reviewer 1 Report

In this work, the complexation of tedizolid with hydroxypropyl-β-cyclodextrin was studied. This complex was studied in detail with a wide range of physicochemical, computational and biological methods, but I think that the manuscript needs to be seriously revised before accepting:

1. The cyclodextrin molecule should not be called a biopolymer. Cyclodextrin polymers look as presented in following works Progress in Polymer Science 2019, 93, 1-35; Photochem. Photobiol. Sci., 2012,11, 1285-1292, etc.

2. I would like the authors to clarify what kind of approach is proposed by the authors of this work, which represents “an opportunity for the drug delivery systems development…”? An approach to the preparation of water-soluble drug form using cyclodextrins has long been proposed.

3. What other tedizolid-based compositions have been obtained previously by other researchers? Can the authors state the advantages of tedizolid/HP-β-CD composition against other tedizolid-based compositions?

4. In the Introduction section, it is necessary to indicate why it is necessary to slow down the release of tedizolid.

5. In the Introduction section, it is also necessary to justify the choice of HP-β-CD. Why did the authors not choose other CDs to complex with tedizolid?

6. The data in Figure 4 must be confirmed by NMR spectroscopy giving more reliable evidence.

7. The data in Table 1 is missing the error value.

8. Information about the stability of tedizolid/HP-β-CD complex should be provided.

9. I would also like to know about the toxicity (and/or hemolysis) of tedizolid in the absence and presence of cyclodextrin in relation to healthy cells.

10. Preparation of tedizolid/HP-β-CD complex should be described clearly. Please add details of complex preparation in section 3.1.

11. Authors should structure the text of whole article, namely, avoid too short paragraphs, consisting of one and two sentences. The one- and two-sentence paragraphs make the manuscript seem too choppy that is generally unacceptable. Authors should improve the structure and style of the article by combining sentences in meaning into larger paragraphs.

Author Response

Manuscript number: ijms-1048531 

Manuscript title: Tedizolid-cyclodextrin system as delayed release drug delivery with antibacterial activity

Author(s): Magdalena Paczkowska-Walendowska, Natalia Rosiak, Ewa Tykarska, Katarzyna Michalska, Anita PÅ‚aziÅ„ska, Wojciech PÅ‚aziÅ„ski, Daria Szymanowska, Judyta Cielecka-Piontek 

Dear Ms. Luiza Barham,

EditorInternational Journal of Molecular Sciences 

We would like to kindly thank You for your and Reviewer’s thorough review that helped us to improve our paper. We have taken into account all the suggestions and have made the necessary changes. Our responses are as follows:  

Comments from the Editors and Reviewer:
-Reviewer 1

In this work, the complexation of tedizolid with hydroxypropyl-β-cyclodextrin was studied. This complex was studied in detail with a wide range of physicochemical, computational and biological methods, but I think that the manuscript needs to be seriously revised before accepting:

  1. The cyclodextrin molecule should not be called a biopolymer. Cyclodextrin polymers look as presented in following works Progress in Polymer Science 2019, 93, 1-35; Photochem. Photobiol. Sci., 2012,11, 1285-1292, etc.

Response: Cyclodextrin is no longer called a biopolymer. The authors apologize for such a mistake. Appropriate corrections have been made throughout the manuscript.

  1. I would like the authors to clarify what kind of approach is proposed by the authors of this work, which represents “an opportunity for the drug delivery systems development…”? An approach to the preparation of water-soluble drug form using cyclodextrins has long been proposed.

Response: The authors fully agree with the Reviewer that preparation of cyclodextrin forms with active pharmaceutical ingridients is a generally known procedure. Nevertheless, the details are always the most important. To date, there are no studies on changes in the physicochemical properties of parent tedizolid (drug with a hydroxymethyl substituent at the C5 position on the oxazolidinone ring)due to the formation of connections with inactive substances such as cyclodextrins. Therefore, the ‘proposed approach’ of combining tedizolid with cyclodextrin addresses the need for an antimicrobial delayed release drug delivery system.

  1. What other tedizolid-based compositions have been obtained previously by other researchers? Can the authors state the advantages of tedizolid/HP-β-CD composition against other tedizolid-based compositions?

Response: To the best of the authors’ knowledge, there are no studies based on tedizolid-polymer compositions. The authors’ research team has previously worked on tedizolid-based delivery systems based on hydroxypropyl methylcellulose (HPMC) and triblock PEO–PPO–PEO copolymers of poly(ethylene oxide) (PEO) and poly(propylene oxide) (PPO) (trade name Pluronic). These studies were added to the manuscript [page 3, lines 117-123].

  1. In the Introduction section, it is necessary to indicate why it is necessary to slow down the release of tedizolid.

Response: The use of cyclodextrin-based systems, in addition to affecting the dissolution rate of biomolecules, may also affect the prolonged and/or delayed release of the active from system.

Tedizolid, as a tedizolid disodium phosphate prodrug (IV administration) or tedizolid phosphate prodrug (orally in the form of coated tablets) is rapidly converted to the bioactive parent tedizolid, through in vivo bioactivation by human phosphatase. The parent tedizolid and its two prodrugs differ significantly in water solubility. The parent tedizolid is sparingly soluble in water with a solubility of 0.136 mg/mL, the solubility of tedizolid phosphate is 0.61 mg/mL, whereas the solubility of disodium phosphate salt is 50 mg/mL. However, in addition to the benefits of prodrugs, we must also bear in mind the limitations of their use. First, the prodrug is inactive prior to metabolism; it becomes active only after biotransformation, which may differ from person to person and depends on the individual metabolic characteristics. Secondly, increasing the amount of phosphate in the human body, administered in the form of a phosphate prodrug, can cause a decrease in calcium levels, which further results in bone weakness, and this will be seriously disadvantageous, especially in the case of tedizolid use in off-label indication for the treatment of bone and joint infections. Therefore, our research concerns the parent tedizolid with low water solubility in order to create a system that will simultaneously improve solubility and have a positive effect on biological membrane permeability while maintaining antimicrobial activity.

Oxazolidinones like linezolid, next beta-lactams, including penicillins and penems, but also glycopeptides, macrolides, azalides and lincosamides are time-dependent antibiotics, meaning that the time that the concentration of a drug remains above the MIC (T > MIC) is the PK-PD index correlating with efficacy. Hence, it is so important to create a delayed release of the active ingredient from the system for tedizolid.

  1. In the Introduction section, it is also necessary to justify the choice of HP-β-CD. Why did the authors not choose other CDs to complex with tedizolid?

Response: HP-βCD was chosen as the optimal solution, due to the European Medicines Agency document (EMA/CHMP/333892/2013) on cyclodextrins used as excipients in medicinal products for human use. HP-βCD are the most versatile cyclic oligosaccharides that can be used in a variety of medicinal products for oral, rectal, dermal, ophthalmic and parenteral administration, and are considered non-toxic if the daily dose is <16 g (270 mg/kg). What is important for implementation a safe pharmaceutical form [page 3, lines 140-146; page 4, lines 152-153].

  1. The data in Figure 4 must be confirmed by NMR spectroscopy giving more reliable evidence.

Response: The main goal of our research was to assess changes in the physicochemical properties of tedizolid, i.e. prolonged release, modification of the penetration through biological membranes or the beneficial change of bactericidal activity, as a result of interaction with the cyclodextrin structure. The identification of the resulting tedizolid-cyclodextrin complex using following methods: XRPD, DSC, FT-IR compared with molecular modelling, is aimed at general characterization of the system and evaluation of major intermolecular interactions. In the authors' opinion, a more precise analysis during responding to reviews of FT-IR spectra and molecular modelling outcomes is sufficient in this respect and carries an exhaustive assessment that the intermolecular interaction between tedizolid and cyclodextrin is responsible for the physicochemical changes in tedizolid behaviour, and not the mere fact of the formation of a physical mixture of those two compounds.

  1. The data in Table 1 is missing the error value.

Response: Error values have been added to Table 1. The authors apologize for such a failure.

  1. Information about the stability of tedizolid/HP-β-CD complex should be provided.

Response: On the basis of the chromatograms analysis before and after the systems preparation, no additional peaks were found, which could derive from the decomposition products of tedizolid due to its kneading. Also during the dissolution and permeation studies, no additional peaks were observed, hence the certainty that no chemical degradation of tedizolid occurred when exposed to solutions with a pH range 1.2-6.8 within 4 hours at 37 oC. Moreover, our conclusions are in line with the literature data on the chemical stability of tedizolid: Michalska et al.: The solution and solid-state degradation study followed by identification of tedizolid related compounds in medicinal product by high performance liquid chromatography with diode array and tandem mass spectrometry detection recently published at Journal of Pharmaceutical and Biomedical Analysis 2020, 113783, where revealed that in solution-state studies, the forced decomposition of tedizolid was stable in oxidative conditions and during thermolysis processes. A tedizolid solution under hydrolytic conditions at room temperature revealed that the sample was very stable with the shelf-life over 5 years as calculated by using an Arrhenius plot (ln A = 18.3). In contrast, in the solid state degradation study, tedizolid was stable under thermal conditions at high humidity and in visible light, while moderate degradation was observed under thermal conditions of low humidity and ultraviolet light. Details on the kinetics of the reaction are available in the article cited above.

Based on author's previous experience, we know that cyclodextrins do not reduce the chemical stability of antibiotics, they can even act as their chemical stabilizers.

  1. I would also like to know about the toxicity (and/or hemolysis) of tedizolid in the absence and presence of cyclodextrin in relation to healthy cells.

Response: All parent cyclodextrins (α-, β- and γ-CD) are accepted as food additives and ‘generally recognized as safe’ (GRAS) [EMA. Cyclodextrins used as excipients. 2017. https://www.ema.europa.eu/en/documents/scientific-guideline/questions-answers-cyclodextrins-used-excipients-medicinal-products-human-use_en.pdf]. As dietary supplement the total daily oral dose of α-CD may reach 6000 mg/day, for β-CD 500 mg/day and for γ-CD 10 000 mg/day, and for HP-β-CD as oral pharmaceutical 8000 mg/day [Loftsson and Brewster: Pharmaceutical applications of cyclodextrins: basic science and product development, Journal Pharmacy and Pharmacology. 2010, 62, 1607-1621]. Moreover, HP-β-CD is referenced in the USP/NF and EP and is cited in the FDA’s list of inactive pharmaceutical ingredients. Based on clinical data, HP-β-CD is considered to be nontoxic if the daily dose is < 16 g (270 mg/kg) [Irie and Uekama. Pharmaceutical applications of cyclodextrins III. Toxicological issues and safety evaluation, Journal of Pharmaceutical Sciences. 1997, 86 (2), 147-162].

                On the other hand, based on tedizolid toxicological report, in the single dose toxicity test, death was observed after dose 2000 mg/kg taken orally in female rats, no death was observed in male rated and mice. In case of repeated dose toxicity test, in the oral studies of 1 and 3 months duration, doses of 100 mg/kg/day were not tolerated [https://www.ema.europa.eu/en/documents/assessment-report/sivextro-epar-public-assessment-report_en.pdf]. Based on these literature data, as well as the authors previous experience, we could conclude that the combination of tedizolid with cyclodextrins is safe and does not poses a toxicological risk.

  1. Preparation of tedizolid/HP-β-CD complex should be described clearly. Please add details of complex preparation in section 3.1.

Response: As requested by the Reviewer more details were added in section 3.1 [page 11, lines 352-356].

  1. Authors should structure the text of whole article, namely, avoid too short paragraphs, consisting of one and two sentences. The one- and two-sentence paragraphs make the manuscript seem too choppy that is generally unacceptable. Authors should improve the structure and style of the article by combining sentences in meaning into larger paragraphs.

Response: Authors appreciate the suggestion to improve our manuscript. We have made many efforts to make an article more readable. We have checked and revised the manuscript carefully according to the Reviewer’s comments.

Reviewer 2 Report

Review on manuscript

 Tedizolid-cyclodextrin system as delayed release 2 drug delivery with antibacterial activity

The submitted manuscript presents a study on tedizolid-cyclodextrin complex obtained by solid dispersion. Cyclodextrins are widely used to enhance the bioactivity of drugs with low water solubility. The study is well composed. In the first part the formation of tedizolid/HP-b-CD inclusion complex is investigated, then the dissolution, permeability and microbiological activity of the new complex is studied.

Despite the techniques used for characterization of tedizolid/HP-b-CD inclusion complex, such as PXRD, DSC, FTIR and molecular docking, the interpretation of the results made me a sense of lack. It is clear for me from the PXRD study that during solid dispersion (kneading) the crystallinity of the drug did not change. The calculation of the entrapped tedizolid from DSC curve, however, was not convincing for me. The intermolecular interactions between the drug and HP-b-CD, proved also by FTIR and molecular docking how is taken into account? Please explain the calculation process.  

As to the FTIR measurements, the presented spectra does not illustrate the wavenumber shifts due to specific interactions. The sentence at row 149 “The FT-IR spectra of tedizolid…” makes no sense. In the next sentence the authors refer their previous article where the IR band assignment for tedizolid was presented. I suggest to briefly present also here for better understanding and convenience. Sentences in rows 170-172 makes no sense. I missed the structure of tedizolid which might enhance the understanding of specific interactions.

The molecular docking part is well presented. However, I am interested what should be the reason for slightly different result obtained using UFF and DFT-derived force fields?

For me is a gap between the outcome of molecular docking calculation and the FTIR spectroscopic part. Please try to interpret better the spectral changes.

The description of the dissolution experiments is confusing. How “powder dissolution” can be understood?  The explanation of the outcome of microbiological test is also a little speculative.

As a general conclusion, I suggest a more thorough analysis of the measurement results. In the present form the manuscript does not contain enough new physical insight to be published in IJMS (IF:4.55). After an amendment of the manuscript, however, could be reconsidered for review and publication.  

Author Response

Manuscript number: ijms-1048531 

Manuscript title: Tedizolid-cyclodextrin system as delayed release drug delivery with antibacterial activity

Author(s): Magdalena Paczkowska-Walendowska, Natalia Rosiak, Ewa Tykarska, Katarzyna Michalska, Anita PÅ‚aziÅ„ska, Wojciech PÅ‚aziÅ„ski, Daria Szymanowska, Judyta Cielecka-Piontek 

Dear Ms. Luiza Barham,

EditorInternational Journal of Molecular Sciences 

We would like to kindly thank You for your and Reviewer’s thorough review that helped us to improve our paper. We have taken into account all the suggestions and have made the necessary changes. Our responses are as follows:

Comments from the Editors and Reviewer:

-Reviewer 2

The submitted manuscript presents a study on tedizolid-cyclodextrin complex obtained by solid dispersion. Cyclodextrins are widely used to enhance the bioactivity of drugs with low water solubility. The study is well composed. In the first part the formation of tedizolid/HP-b-CD inclusion complex is investigated, then the dissolution, permeability and microbiological activity of the new complex is studied.

Despite the techniques used for characterization of tedizolid/HP-b-CD inclusion complex, such as PXRD, DSC, FTIR and molecular docking, the interpretation of the results made me a sense of lack. It is clear for me from the PXRD study that during solid dispersion (kneading) the crystallinity of the drug did not change. The calculation of the entrapped tedizolid from DSC curve, however, was not convincing for me. The intermolecular interactions between the drug and HP-b-CD, proved also by FTIR and molecular docking how is taken into account? Please explain the calculation process. 

Response: In accordance with the valuable remark of the Reviewer, the methods of identifying the newly created system have several applications. As we already known, HP-β-CD occurs in an amorphous form, therefore the applied PXRD was to confirm the crystallinity or the transition of tedizolid into amorphous state. The binary system diffractogram is a sum of patterns of each individual compounds (tedizolid and HP-β-CD).

The DSC thermograms gave us information about enthalpy of fusion. Based on the knowledge and experience of the team of prof. Marian Paluch from the Institute of Physics, University of Silesia, for the first time we proposed the use of the enthalpy of fusion model for cyclodextrin-based systems in 2018 (Paczkowska et al.: Enhanced pharmacological efficacy of sumatriptan due to modification of its physicochemical properties by inclusion in selected cyclodextrins. Scientific Reports 2018, 8, 16184). Since that, we are using a simple mathematical equation to describe further research. Based on results of presented work, please look on below-given explanation:

  1. ΔHTED = 120.3215 J/g
  2. ΔHTED/HP-β-CD = 20.0647 J/g
  3. %TED in TED/HPβCD à
  4. Under ideal conditions, ΔHTED/HP-β-CD would be à32 x 20.22% = 24.33 J/g
  5. But the experimental ΔHTED/HP-β-CD is different, therefore the % inclusions can be calculated à

Intermolecular interactions between the tedizolid and HP-β-CD have been concluded based on FT-IR spectra and molecular docking – please see below answer.

As to the FTIR measurements, the presented spectra does not illustrate the wavenumber shifts due to specific interactions. The sentence at row 149 “The FT-IR spectra of tedizolid…” makes no sense. In the next sentence the authors refer their previous article where the IR band assignment for tedizolid was presented. I suggest to briefly present also here for better understanding and convenience. Sentences in rows 170-172 makes no sense. I missed the structure of tedizolid which might enhance the understanding of specific interactions.

Response: Wavenumber shifts were added to FT-IR spectra as well as tedizolid structure [Figure 3]. The previous outcomes are presented in revised version of manuscript [page 5, lines 190-196].

The molecular docking part is well presented. However, I am interested what should be the reason for slightly different result obtained using UFF and DFT-derived force fields?

Response: At the most basic level, it can be explained by the inherent differences between predictions of both UFF and DFT potentials. For the structural reasons, the different results originate from the high conformational flexibility of the 2-HP substituents, comparing to the rest of the molecule. The mentioned differences involve mainly the exocyclic, HP substituents, which are the most flexible part of the molecule. Those substituents contain 5 rotatable bonds and each of them (even when considering only the three staggered rotamers per one rotatable bond) can exhibit 243 different conformations. Each of the tested potentials predict slightly different favourable conformation of the substituent. Deducing from the conformational properties of the involved –C-C-, -C-O- and –O-H bonds, it can be stated that in reality the orientation of the substituents is not ‘frozen’ but can adopt various geometries, undergoing the dynamical exchange within several picoseconds. This is probably correlated with the coexistence of multiple different binding modes, differing mainly by the pattern of interactions of the ligand molecule with 2-HP moieties. Thus, we preferred to consider the two alternative conformations of the whole HP-CD molecule, instead restricting to the single one, origination from more accurate, DFT potential. As expected, the found binding modes differ mainly with respect to the interactions involving HP groups. The two sentences have been added to the revised version of the manuscript, summarizing the explanation given above [page 8, lines 249-256].

For me is a gap between the outcome of molecular docking calculation and the FTIR spectroscopic part. Please try to interpret better the spectral changes.

Response: A deeper interpretation of the FT-IR spectrum along with a comparison to the molecular modelling results was carried out. Extended interpretation has been added to revised version of manuscript [pages 5-6, lines 211-226].

The description of the dissolution experiments is confusing. How “powder dissolution” can be understood?  The explanation of the outcome of microbiological test is also a little speculative.

Response: In our dissolution experiment, we wanted to alter the changes in the tedizolid dissolution rate profiles. Dissolution profiles are mainly obtained for solid dosage forms e.g. tablets/capsules. There is more and more literature data about dissolution rate profiles of powder systems, not compressed ones. So that’s why Authors also use such a modified dissolution method to obtain changes in chemical properties of cyclodextrin system. If we look into experiment description, it is based on pharmacopeia methodology for dosage forms, with the difference that the powder system is placed in a gelatin capsule as a vehicle for placement in the acceptor medium. Gelatin capsule is not considered in that case as pharmaceutical dosage form. This approach to testing the dissolution rate of powder systems has been described in many previous works: Paczkowska et al.: Mechanochemical activation with cyclodextrins followed by compaction as an approach to improve dissolution of rutin, International Journal of Pharmaceutics 2020, 581, 119294, doi: 10.1016/j.ijpharm.2020.119294; Paczkowska et al.: influence of inclusion into β-cyclodextrin on physicochemical and biological properties of tebipenem pivoxil. Plos One 2019, https://doi.org/10.1371/journal.pone.0210694; Paczkowska et al.: The analysis of the physicochemical properties of benzocaine polymorphs, Molecules 2018, 23(7), 1737, doi: 10.3390/molecules23071737; Paczkowska et al.: β-cyclodextrin complexation as an effective drug delivery system for meropenem, European Journal of Pharmaceutics and Biopharmaceutics 2016, 99: 24-34, doi: 10.1016/j.ejpb.2015.10.013; Paczkowska et al.: Complex of rutin with β-cyclodextrinas potential delivery system. Plos one 2015, https://doi.org/10.1371/journal.pone.0120858. However, accepting the comments of the Reviewer, the description of figure 5 has been changed to avoid confusing ‘powder dissolution’ term.

A microbiological analysis to determine the minimum inhibitory concentration (MIC) of tedizolid (control) and the active substance enriched with an additive such as HP-β-CD has been performed for bacterial strains responsible for both general infections and acute bacterial skin and skin structure infections (ABSSSI). The results of our research confirm previous ones described in Cyclodextrins as multifunctional excipients: Influence of inclusion into β-cyclodextrin on physicochemical and biological properties of tebipenem pivoxil. PLoS One. 2019, 14(1), https://doi.org/10.1371/journal.pone.0210694. Taking into account a double decrease of MICs characterising the antibacterial activity of the tedizolid/HP-β-CD complex in relation to Enterococcus faecalis ATTC 29212 and even fourfold decrease in case of clinical isolates, it seems possible to suggest mechanism inhibiting a development of selected strains of the above-mentioned bacteria in the presence of the complex. So, blocking porin channels can contribute to the efflux effect in bacteria by cyclodextrins, and destabilization of the tight junction proteins can simultaneously inhibit the efflux transporters, what was confirmed in previous literature data [Arima et al., Contribution of P-glycoprotein to the enhancing effects of dimethyl-beta-cyclodextrin on oral bioavailability of tacrolimus, J Pharmacol Exp Ther. 2001, 297(2), 547-555; Haimhoffer et al., Cyclodextrins in Drug Delivery Systems and Their Effects on Biological Barriers, Scientia Pharmaceutica 2019, 87, 33].

As a general conclusion, I suggest a more thorough analysis of the measurement results. In the present form the manuscript does not contain enough new physical insight to be published in IJMS (IF:4.55). After an amendment of the manuscript, however, could be reconsidered for review and publication. 

Response: We appreciate the suggestion to improve our manuscript. We have made many efforts to make an article more readable. We have checked and revised the manuscript carefully according to the Editor’s and the Reviewer’s comments.

Round 2

Reviewer 1 Report

With reference to the previously submitted manuscript ijms-1048531, authors have satisfactorily addressed most of the points raised by the reviewer. Manuscript has been substantially revised accordingly. The present version of the manuscript may be accepted for publication in IJMS.